# *PtrABR1* Increases Tolerance to Drought Stress by Enhancing Lateral Root Formation in *Populus trichocarpa*

**DOI:** 10.3390/ijms241813748

**Published:** 2023-09-06

**Authors:** Lijiao Sun, Xinxin Dong, Xingshun Song

**Affiliations:** 1State Key Laboratory of Tree Genetics and Breeding, Northeast Forestry University, Harbin 150040, China; sunlj330@163.com (L.S.); xdtxzka@163.com (X.D.); 2College of Life Science, Northeast Forestry University, Harbin 150040, China

**Keywords:** *Populus trichocarpa*, *PtrABR1*, PtrYY1, lateral root, drought tolerance

## Abstract

Roots are the main organ for water uptake and the earliest part of a plant’s response to drought, making them of great importance to our understanding of the root system’s response to drought. However, little is known about the underlying molecular mechanisms that control root responses to drought stress. Here, we identified and functionally characterized the AP2/ERF family transcription factor (TF) *PtrABR1* and the upstream target gene zinc-finger protein TF PtrYY1, which respond to drought stress by promoting the growth and development of lateral roots in *Populus trichocarpa*. A root-specific induction of *PtrABR1* under drought stress was explored. The overexpression of *PtrABR1* (*PtrABR1-OE*) promoted root growth and development, thereby increasing tolerance to drought stress. In addition, PtrYY1 is directly bound to the promoter of *PtrABR1* under drought stress, and the overexpression of *PtrYY1* (*PtrYY1-OE*) promoted lateral root growth and development and increased tolerance to drought stress. An RNA-seq analysis of *PtrABR1-OE* with wild-type (WT) poplar identified *PtrGH3.6* and *PtrPP2C44*, which share the same pattern of expression changes as *PtrABR1*. A qRT-PCR and cis-element analysis further suggested that *PtrGH3.6* and *PtrPP2C44* may act as potential downstream targets of PtrABR1 genes in the root response pathway to drought stress. In conclusion, these results reveal a novel drought regulatory pathway in which *PtrABR1* regulates the network through the upstream target gene PtrYY1 and the potential downstream target genes *PtrGH3.6* and *PtrPP2C44*, thereby promoting root growth and development and improving tolerance to drought stress.

## 1. Introduction

During their growth and development, plants are subject to a variety of environmental pressures, with drought being one of the most deleterious. Drought severely limits plant productivity and poses a threat to global food security and sustainable agriculture [1,2]. To cope with water-scarce environments, plants have evolved intrinsic mechanisms and adaptive strategies. One of these key strategies is to optimize the structural characteristics of the root system [3,4,5,6]. Root architecture plays a crucial role in determining water use efficiency since plants absorb water through their roots [7,8,9]. Understanding the phenotypic plasticity of root architecture and the potential genetic basis of these responses in variable soil environments is crucial for enhancing drought adaptation by combining root-related traits [10]. Therefore, there has been a growing interest in exploring the relationship between root architecture and drought tolerance [10,11,12]. Lateral roots are a major component of the plant root system, and the developmental status of lateral roots is closely related to the drought resistance of plants [13]. Increasing the number of lateral roots can absorb more water in the soil and thus withstand drought in plants at the seedling stage [14]. In recent years, a significant corpus of literature has emerged that investigates the impact of lateral roots on drought stress. For example, the overexpression of chrysanthemum *CmERF053* in Arabidopsis promoted the growth of primary and lateral roots and enhanced drought tolerance [15]. Arabidopsis *AtMYB77* was involved in drought-induced lateral root growth by promoting NO synthesis [16]. Compared with wild-type seedlings, maize *ZmLRT* knockout lines exhibited a significant increase in lateral roots at the seedling stage and improved drought tolerance [17]. The overexpression of *MsDREBA6* in *Malus sieversii* plants in Arabidopsis increased the lateral root length and density and improved drought tolerance [18]. In addition, PtabZIP1L, homologous to Arabidopsis bZIP1, has been reported to be a positive regulator of lateral root growth and drought tolerance in poplar [19]. The overexpression of *ERF016* in 84K poplar (*Populus alba* × *P. glandulosa*) enhanced root drought tolerance by increasing the number of lateral roots and other characteristics [20].

Plants have evolved intricate mechanisms to survive in harsh environmental conditions and maintain growth and development. One such mechanism involves the transcriptional reprogramming of stress-responsive molecular modules, which consist of transcription factors (TFs) and their associated target genes. One of the largest families of TFs in plants, the AP2/ERF(APETALA2, Ethylene-responsive element binding factor) superfamily, has one or two AP2 structural domains with 60–70 conserved amino acid residues [21]. Based on sequence similarity and the number of AP2/ERF domains, the superfamily was further subdivided into four families: ERF, AP2, RAV (linked to ABI3/VP), and Soloist [22,23]. Among them, the ERF subfamily has a single AP2 domain with 60 conserved amino acids that are responsible for DNA binding [24]. In addition, scant evidence indicates that ABR1 (AP2-like abscisic acid repressor 1), an ERF family gene, plays a significant role in plant responses to abiotic stresses. For example, Arabidopsis AtABR1 acts as a blocker of ABA responses during seed germination and stress-induced gene expression [25], resulting in a distinct phenotype of root hair length and number [26]. The overexpression of *BjABR1* in *Brassica juncea* attenuates its susceptibility to ABA, salt, and osmotic stress [27]. ABR1 in *Sesuvium portulacastrum* has been shown to be involved in the salt stress pathway [28]. PtrABR1 and PtrABF4 in *Poncirus trifoliata* positively regulate plant drought tolerance by controlling soluble sugar accumulation from BAM3-mediated starch breakdown [29]. The above highlights the need to elucidate the regulation of ABR1 genes in response to abiotic stress. However, the upstream regulatory factors of ABR1 remain poorly understood, particularly in perennial plants.

It is well known that phytohormones have a wide range of roles in the transduction of stress signals that trigger physiological and metabolic changes in plants. Abscisic acid (ABA) is the most characterized and instrumental messenger that regulates drought stress response [30]. AtYY1 (YIN YANG 1) and ABR1 have been shown to be the key factors of the ABA signaling pathway, which plays an important role in the regulation of abiotic stresses [31]. For example, Arabidopsis *AtYY1* is involved in plant drought resistance and ABA signaling [1]. *ABR1*, as a repressor of the ABA signaling pathway, is considered to be the main target of AtYY1 to regulate the ABA signaling pathway [31]. Maize ZmYY1, a homologous protein of AtYY1, positively regulates the expression of *ZmABR1*, which further modulates the ABA signaling pathway as well as improves tolerance to salt and drought stress [32]. PtrABR1 in *Poncirus trifoliata* was significantly induced by drought and ABA treatments as well as by regulating PtrBAM3-mediated starch degradation and sugar accumulation to enhance drought tolerance [29]. Considering the role of ABRs in ABA signaling and the extensive role of ABA in drought stress, it is of great interest to explore the mechanism of ABR1-mediated drought tolerance in woody plants.

*Populus*, a genus of woody perennials, comprises fast-growing species that have been extensively utilized as a model system for investigating the physiology and genetics of angiosperms [33]. However, severe environmental drought stress limits the water metabolism, stomatal structure, and osmoregulatory balance in poplar. Therefore, there is a need to explore drought tolerance genes, exploit their regulatory mechanisms, and breed drought-tolerant poplar species [34]. In this study, we describe the role of *PtrABR1*, an ERF family gene with an AP2 structural domain, in *P. trichocarpa*. The *PtrABR1* gene is highly specifically expressed in roots and is directly induced by the zinc-finger protein TF PtrYY1, thereby improving tolerance to drought stress by promoting lateral root growth and development. In addition, PtrABR1 potentially inhibits the expression of *GH3.6* and *PP2C44* to improve drought tolerance in poplar, and PtrABR1-mediated drought tolerance contributes to the development of drought-tolerant poplar with a high economic and environmental value. It also contributes to our present understanding of the molecular mechanisms involved in the response of poplar roots to drought.

## 2. Results

### 2.1. PtrABR1 Expression Is Significantly Induced by Drought Stress

By analyzing RNA-seq data (accession number GSE81048 [35]) on drought stress in *P. trichocarpa*, we identified *PtrABR1* TF (*Potri.002G065600*) from the AP2/ERF family that may respond to drought stress. To evaluate whether *PtrABR1* plays a role in abiotic stress, we subjected 20-day-old *P. trichocarpa* seedlings to drought, ABA, NaCl, and low-temperature stresses, respectively.

We found that drought, salt, and ABA stresses all significantly induced the expression of *PtrABR1*, especially drought stress, which resulted in over 100 times higher levels of the expression of *PtrABR1* after 3 h of treatment compared to the untreated samples. In contrast, the low-temperature treatment significantly reduced the expression of *PtrABR1* (Figure 1A–D). A tissue expression analysis of *PtrABR1* revealed that it was expressed in roots, stems, and leaves, with the highest expression level observed in the roots, followed by leaves (Figure 1E). Additionally, we cloned a 1980 bp fragment upstream of the start codon into the PBI121-*GUS* vector and successfully transformed four transgenic plants of *P. trichocarpa* (Appendix A). PtrABR1pro::*GUS* transgenic plants showed darker GUS staining in roots at 20 day compared to that at 10 day (Figure 1F); it was speculated that *PtrABR1* was involved in root growth and development. In summary, the *PtrABR1* gene responds to drought stress and plays an important role in root growth and development.

### 2.2. Characterization of the PtrABR1 Gene in P. trichocarpa

Multiple sequence comparisons showed that PtrABR1 shares a conserved structural domain of 60 bp with ABR1 from other species, consistent with the structural features of AP2/ERF family proteins [36]. Notably, the residues at the 14th and 19th sites in the structural domain of PtrABR1 are alanine and aspartate, respectively (Figure 2A), which are characteristic of the ERF subfamily [37], suggesting that AP2/ERF is classified in the ERF subfamily. To investigate the subcellular localization of the PtrABR1 protein, the coding sequence (CDS) of PtrABR1 was fused into a PBI121 vector containing green fluorescent protein (GFP). It was found that the green fluorescent protein of the empty vector (35S: GFP) was distributed throughout the cell, and PtrABR1 (35S: PtrABR1-*GFP*) was distributed in the tobacco cell membrane and cytoplasm. In addition to this, 35S: PtrABR1-GFP could not overlap with chloroplast fluorescence, ruling out the possibility that it was located in chloroplast organelles (Figure 2B).

To assess the transcriptional activation activity of PtrABR1, full-length and truncated CDS sequences were fused to the GAL4-binding domain (BD) in the pGBKT7 vector and subsequently transformed into yeast. The results showed that only the full-length and N-terminal PtrABR1 could grow in SD/-Trp/-His and produce a blue spot in SD/-Trp/-His/X-α-gal medium (Figure 2C), as they were transcriptionally active transformants capable of activating the expression of the reporter gene His. Furthermore, only when the GAL4 gene is activated, α-galactosidase is secreted and hydrolyzes X-α-gal in the medium, causing the colony to produce a blue spot (Figure 2C). These results indicate that PtrABR1 is a transcription activator, and its transactivation domain is located at the N-terminus, excluding the AP2 domain.

### 2.3. PtrABR1 Positively Regulates Lateral Root Tolerance under Drought Stress

We obtained eight overexpression plants (*PtrABR1-OE*) of *P. trichocarpa* via an *Agrobacterium* infection, and their transgenic status was analyzed and confirmed via PCR and RT-qPCR testing (Appendix A). Three OE lines with the highest expression levels, *PtrABR1-OE2*, *PtrABR1-OE6*, and *PtrABR1-OE7*, were selected for further analysis. Under normal growth conditions, the root phenotypes of the transgenic plants were significantly different from those of the WT plants (Figure 3A), with the number of lateral roots being significantly higher in *PtrABR1-OE* than that in WT plants (Figure 3B). In addition, the dry weight of roots in transgenic plants was also significantly higher than that in WT plants (Figure 3C), indicating that *PtrABR1* promoted the formation of lateral roots.

The high permeability of polyethylene glycol PEG6000 simulates drought stress, and experimental conditions are easy to control using a short cycle [38]. After 7 day of growth under normal rooting medium followed by 14 day of growth under 6% PEG6000 rooting medium, it was found that the growth of the root system was significantly inhibited in both *PtrABR1-OE* and WT plants (Figure 3), but the number of lateral roots, dry weight of the roots, and height of the plant were significantly higher in *PtrABR1-OE* than in the WT plants (Figure 3B,C and Appendix A). The lateral root system constitutes the bulk of the root architecture [39] and may absorb most of the water [40]. Therefore, we hypothesized that *PtrABR1-OE* may have improved water utilization by promoting the growth of lateral roots, thereby increasing drought tolerance. Malondialdehyde (MDA) and electrolyte leakage (EL) are typical indicators of cellular damage due to lipid peroxidation and membrane damage, respectively [41]. The results showed that the MDA content and EL in the roots of *PtrABR1-OE* plants were significantly lower than those of WT plants under drought stress (Figure 3D,E), suggesting that *PtrABR1* improved their tolerance to drought stress. These results suggest that *PtrABR1* may improve plants’ tolerance to drought stress by promoting lateral root growth.

To further investigate the role of *PtrABR1* in enhancing drought tolerance through roots, 50-day-old potted plants were subjected to drought for 14 day followed by 5 day of re-watering (Figure 4A). Although there was no significant difference in plant height between the *PtrABR1-OE* and WT plants (Figure 4A,B), the survival rate of the transgenic plants was significantly higher than that of the WT plants after 5 day of water resumption (Figure 4C). Root morphological characteristics are positively correlated with plant drought resistance [11,12,42], and the lateral root density helps improve the response of wheat to water stress and alter yield composition [43]. The roots of *PtrABR1-OE* plants were generally stronger with dense lateral roots (Figure 4A), and their dry and fresh weights were significantly higher than those of WT plants (Figure 4D,E). After the drought treatment, the levels of EL and MDA in the roots of the *PtrABR1-OE* plants were significantly lower than those of the WT plants, and the proline (Pro) content was significantly higher than that of the WT plants (Figure 4F–H), indicating that *PtrABR1* improved the roots’ tolerance to drought stress.

### 2.4. PtrYY1 and PtrSPL10 Bind to the PtrABR1 Promoter

Arabidopsis zinc-finger TFs Yin Yang 1 (AtYY1) and squamosa promoter binding protein-like 10 (SPL10) can bind to the *ABR1* promoter and directly regulate *ABR1* expression, while the encoding AP2 domain TF ABSCISIC ACID INSENSIVE4 (ABI4) can antagonize this regulation [31,44]. In addition, YY1, ABI4, and SPL10 have similar functions to PtrABR1 and are involved in the regulation of lateral root development and drought resistance [45,46]. Therefore, we hypothesized that PtrABR1 may function similarly to Arabidopsis ABR1 and be directly regulated by *PtrYY1* (*Potri.004G221700*), *PtrSPL10* (*Potri.001G055900*), and *PtrABI4* (*Potri.008G071100*) to regulate lateral root development and drought resistance. In this study, under drought stress, *PtrYY1*, *PtrSPL10*, and *PtrABI4* were significantly induced in the roots of *P. trichocarpa* (Figure 5A–C). The results suggest that these factors function similarly to *PtrABR1* and may regulate lateral root development and drought resistance by binding directly to the promoter of *PtrABR1*.

To determine whether *PtrABR1* is directly regulated by PtrYY1, PtrSPL10, and PtrABI4, we performed Y1H dual luciferase assays and histochemical GUS staining. The promoter sequence of *PtrABR1* was able to interact with PtrYY1 and PtrSPL10, respectively, in an SD/-Leu medium containing 300 ng/mL AbA (Figure 6A and Appendix A). Histochemical GUS staining using the *PtrABR1* promoter fused to the GUS reporter gene showed that PtrYY1 and PtrSPL10 deepened and diminished the blue color, respectively, compared to the empty vector (pGreenII 62-SK), and an analysis of the relative expression of GUS showed the same trend (Figure 6B–D). Furthermore, using a dual luciferase assay, PtrYY1 significantly increased transcriptional activation of the *PtrABR1* promoter, while PtrSPL10 showed the opposite trend, repressing *PtrABR1* promoter activity (Figure 6E,F) In conclusion, the results indicate that PtrYY1 and PtrSPL10 can bind directly to the *PtrABR1* promoter and activate and repress *PtrABR1* promoter activity, respectively.

### 2.5. PtrYY1 Positively Regulates Lateral Root Tolerance under Drought Stress

We obtained four overexpression lines (*PtrYY1-OE*) in *P. trichocarpa* through an Agrobacterium infection and confirmed their transgenic status via PCR and RT-qPCR analyses (Appendix A). We selected three *PtrYY1-OE* lines, *PtrYY1-OE1*, *PtrYY1-OE2*, and *PtrYY1-OE4*, with the highest expression levels for further analysis. Under normal growth conditions, there were significant differences in root phenotypes (Figure 7A), with *PtrYY1-OE* having significantly more lateral roots than WT plants (Figure 7B). Additionally, the transgenic plants had a significantly higher root dry weight than the WT plants (Figure 7C), indicating that *PtrYY1* regulates lateral root development in *P. trichocarpa*.

When grown under normal rooting medium for 7 day followed by 6% PEG6000 rooting medium for 14 day, the root growth was found to be significantly inhibited in both *PtrYY1-OE* and WT plants, but the number of lateral roots, root dry weight, and plant height were significantly higher in *PtrYY1-OE* than in WT plants (Figure 7A–C and Appendix A). Therefore, we hypothesized that *PtrYY1-OE* may have improved water utilization by promoting the growth of lateral roots, thereby enhancing drought tolerance. In addition, the MDA and EL levels in the roots of the *PtrYY1-OE* plants were significantly lower than those of the WT plants (Figure 7D,E), suggesting that *PtrYY1* improved their tolerance to drought stress. In short, these results suggest that *PtrYY1* may improve drought stress tolerance in *P. trichocarpa* by promoting lateral root growth and increasing water utilization.

Next, we subjected 50-day-old *PtrYY1-OE* and WT soil-cultivated seedlings to a drought treatment for 14 day, followed by rehydration for 5 day (Figure 8). Before the stress treatment, the plant height of the *PtrYY1-OE* plants was found to be significantly lower than that of the WT plants, and it was hypothesized that *PtrYY1* may play an important role in plant growth and development. In addition, we determined the MDA, EL, and Pro content before the treatment and found no significant differences, suggesting that they were in the same growth state and were not stressed at that time (Figure 8F–H). After the drought stress treatment, the survival rate of *PtrYY1-OE* was found to be significantly lower than that of the WT plants (Figure 8A,B), whereas, after 5 day of rehydration, the survival rate of the transgenic plants was significantly higher than that of the WT plants (Figure 8C). By observing the phenotypes, it was found that the *PtrYY1-OE* plants generally had a stronger root system with denser lateral roots (Figure 8A), and their dry and fresh weights were significantly higher than those of the WT plants (Figure 8D,E). In addition, the EL and MDA levels in the roots of the *PtrYY1-OE* plants were significantly lower and the Pro content was significantly higher than that of the WT plants after the drought treatment (Figure 8F–H). Based on these findings, it can be inferred that *PtrYY1* plays a key role in drought stress response and may enhance plant drought tolerance by promoting lateral root growth during droughts.

### 2.6. Transcriptome Analysis of PtrABR1 Transgenic Plants and WT Plants

To gain further insight into the molecular mechanisms of *PtrABR1* in roots and to identify potential target genes regulated by *PtrABR1*, we performed an RNA-seq analysis using root tissues from transgenic (T) and WT (C) poplar. A total of 347 genes showed altered transcript levels compared to those of the WT poplar (fold change ≥ 2, FDR < 0.05), with 194 genes being up-regulated and 153 genes being down-regulated (Figure 9A,B).

The gene ontology (GO) analysis indicated that DEGs were enriched in biological processes, molecular functions, and cellular component pathways, particularly in cell part and catalytic activity processes (Figure 9C). In response to abiotic stresses such as drought, enzymatic and catalytic activities in plants, such as superoxide dismutase, are altered, which may lead to increased oxidative stress and affect the organism’s growth and development. Eight genes were selected for the validation of the transcriptome analysis’ results (Figure 9D), among which *GH3.1* (*Potri.001G298300*), *GH3.6* (*Potri.014G136800*), *MADS-box18* (*Potri.003G170000*), and *MADS-box27* (*Potri.009G079000*) may be involved in root growth and development [47,48], and *RGLG2* (*Potri.006G002600*), *PP2C44* (*Potri.008G168400*), *MYB4* (*Potri.002G038500*), and *ASR3* (*Potri.005G193600*) may play important roles in abiotic stress [49,50,51,52]. The results of the qRT-PCR analysis were highly consistent with the RNA-Seq expression trends, indicating that the DEGs identified through the RNA-Seq screening were reliable.

### 2.7. Prediction of Downstream Target Genes of PtrABR1 under Drought Stress

Drought regulation through lateral roots is accomplished by the involvement of multiple response genes. Previous research has shown that growth hormone is the major plant hormone regulating lateral root development at every stage of the process [13], with the IAA-coupled gene *GH3.6* playing a critical role in this process [47]. For example, MdGH3.6 was found to inhibit root development and negatively regulate drought [47,53]. Additionally, protein phosphatases type 2C (PP2Cs), a key negative regulator of ABA signaling, negatively regulates root growth and water solubility responses [54]. To investigate whether PtrABR1 mediates the expression of the drought-responsive genes, we identified the genes *GH3.6* and *PP2C44*, whose expression trends were consistent with the expression trend of the PtrABR1 gene in WT and *PtrABR1-OE* plants according to our RNA-seq results (Figure 10A–C). These findings suggest that *GH3.6* and *PP2C44* might be downstream target genes of PtrABR1.

To verify whether *GH3.6* and *PP2C44* function as potential downstream target genes of PtrABR1 in the drought pathway of roots, we selected WT and *PtrABR1-OE* plants that had been grown in histologic flasks for 20 day and exposed them to drought conditions on filter paper. We then selected roots for our qRT-PCR analysis (Figure 10D,E). The expression trends of *GH3.6* and *PP2C44* further verified that they could be downstream target genes of PtrABR1 and that their expression was negatively regulated by PtrABR1 to promote root development and drought resistance.

The AP2/ERF family of genes contains an AP2 structural domain that can bind directly to DRE/CRT cis-acting elements or GCC box cis-acting elements, which are commonly found in the promoter regions of many stress response and tolerance genes [37,55]. Previous studies have also shown that *PtrABR1* in *Poncirus trifoliata* acts as a transcriptional activator of *PtrBAM3* by binding to the GCC box [29]. An analysis of the cis-acting elements in the 2000 bp promoter sequences of *GH3.6* and *PP2C44* revealed that these promoters contain the GCC box and DRE cis-acting elements, respectively (Appendix A). This further supports the notion that *GH3.6* and *PP2C44* may be downstream target genes of PtrABR1.

## 3. Discussion

Previous studies have shown that ERF proteins enhance resistance to various abiotic stresses, including drought [56], salt [57], cold [58], and fungal disease [59]. However, there are few studies on the involvement of ERF genes in the root system’s response to abiotic stresses. Here, we established that the *PtrABR1* TF plays a crucial role in enhancing drought tolerance in *P. trichocarpa*. The overexpression of *PtrABR1* in plants promoted lateral root growth and development and improved drought tolerance. In addition, *PtrABR1* was directly induced by PtrYY1 and potentially regulated *GH3.6* and *PP2C44* expression to promote lateral root growth and development and enhance drought tolerance (Figure 11).

### 3.1. PtrABR1 Enhances Drought Tolerance by Promoting Lateral Root Growth and Development

Optimizing the structural characteristics of the root system is an important and emerging approach to improving plants’ adaptation to drought [4,5,6]. Understanding the phenotypic plasticity of root architecture and its potential genetic basis in response to diverse soil environments is crucial for combining root-related traits to improve drought adaptations [10]. Altered root architecture consists of primary and lateral roots, and the formation of lateral roots is a key feature in exploring the soil in search of nutrients and water [60,61,62]. In this study, it was found that *PtrABR1* was the most responsive to drought stress among different stress treatments. The tissue expression analysis and promoter GUS staining of *PtrABR1* showed that it is mainly expressed in roots and plays an important role in regulating root growth and development (Figure 1). Furthermore, *PtrABR1-OE* plants exhibited a considerably higher lateral root number, root fresh weight, and root dry weight than WT plants when grown in a 6% PEG6000 rooting medium (Figure 3), indicating that *PtrABR1-OE* plants facilitate the development of lateral roots and contribute significantly to the growth and development of the root system. In dicotyledons, lateral root formation plays a significant role in enhancing the ability of the root system to absorb water and nutrients, as well as anchor the plant in the soil. Previous studies have illustrated that Arabidopsis *HDG11* mutants exhibit increased lateral root number and improved tolerance to drought stress compared to WT plants [63]. Root morphological characteristics have been observed to be positively associated with plant drought tolerance [11,12,42]. Additionally, increasing lateral root density may help improve wheat’s response to water stress and alter its yield composition [43]. Furthermore, the roots of *PtrABR1-OE* plants exhibited lower MDA and EL contents than those of WT plants (Figure 3). Similar results were obtained after a drought treatment of 50-day-old potted plants (Figure 4). These findings suggest that *PtrABR1* might enhance tolerance to drought stress by promoting lateral root growth.

### 3.2. PtrYY1 Promotes Lateral Root Growth by Directly Regulating the Expression of PtrABR1, Thereby Enhancing Drought Resistance

Previous studies have demonstrated that Arabidopsis AtYY1 can bind to the *ABR1* promoter and directly up-regulate *ABR1* expression in response to abiotic stress [31]. Moreover, SPL10 can bind to the *ABR1* promoter sequence to regulate root growth and development [64]. However, ABI4, an AP2 domain TF, can counteract the regulation of ABR1 by AtYY1 and is involved in regulating lateral root development and drought resistance [31,45,46]. To explore whether a similar regulation is present in poplar, we performed yeast one-hybrid (Y1H) assays, histochemical GUS staining, and dual luciferase reporter gene assays. It was shown that PtrYY1 and PtrSPL10 act as upstream genes directly regulating the expression of *PtrABR1* (Figure 6). YY1 acts as a bifunctional TF, both as an activator and a repressor [31,65]. YY1 was first identified in Drosophila and mammals [65,66,67] and is evolutionarily conserved between the animal and plant kingdoms [31]. YY1, a member of the GLI Krüppel family, contains four Cys2-His2 (C2H2) zinc fingers that form structural domains for DNA binding [65]. To date, studies on YY1 in plants are scarce, and it has only been reported in a few species. The overexpression of *AtYY1* in Arabidopsis leads to enhanced root growth and plays an important role in abiotic stresses [31]. In addition, in recent studies, the expression of two *AtYY1* paralogs of *P. trichocarpa*, *YIN* and *YANG* (known as *PtYY1a* and *PtYY1b*), enhanced root growth in Arabidopsis [68]. In our study, *PtrYY1-OE* plants promoted lateral root growth and development and increased tolerance to drought stress (Figure 7), indicating that PtrYY1 directly regulates the expression of *PtrABR1* to improve drought resistance by promoting lateral root growth; we obtained similar results to those of previous authors.

The plant hormone ABA plays an important role in drought stress [30], and it has been reported that ABR1 and YY1 are ABA-signaling key factors [31]. Arabidopsis AtYY1 directly binds to *ABR1* to negatively regulate the ABA signaling pathway and limit drought tolerance in plants [31]. Interestingly, *PtrABR1* in *Poncirus trifoliata* was significantly induced by drought and ABA treatments and functioned as a positive regulator in drought stress. Our results are similar to previous findings that *PtrABR1* is a target of PtrYY1 that positively regulates drought tolerance [29]. Considering that *PtrABR1* was significantly induced under ABA treatment (Figure 1C), it is speculated that *PtrABR1*, as a target gene of PtrYY1, may also regulate drought resistance by participating in ABA signaling. It should be noted that the specific molecular mechanisms of *PtrYY1* and *PtrABR1* involved in the ABA signaling pathway to regulate drought tolerance in *P. trichocarpa* need to be further investigated.

### 3.3. PtrABR1 Is Predicted to Directly Regulate the Expression of GH3.6 and PP2C44 to Improve Tolerance to Drought Stress

In this study, we identified two DEGs, *GH3.6* and *PP2C44*, from the roots of *PtrABR1-OE* and WT plants by analyzing the RNA-Seq results (Figure 9). It was observed that the expression trends of *GH3.6* and *PP2C44* in WT plants and *PtrABR1-OE* were consistent with those of PtrABR1 in different lines and were responsive to drought stress (Figure 9). Therefore, it is possible that PtrABR1 directly inhibits the expression of *GH3.6* and *PP2C44* to improve drought resistance. Plant growth hormone is a crucial regulator of lateral root development at every stage [13], and GH3.6, an IAA-coupled gene, has been implicated in regulating root development [47]. The transcription of GH3.6 is repressed by WRKY46, which binds directly to its promoter, thereby maintaining free indole-3-acetic acid (IAA) content and root growth [47]. Additionally, *MdGH3.6* has been shown to inhibit root development and negatively regulate drought [47]. PP2Cs, which are negative regulators of ABA signaling, have been demonstrated to negatively regulate root growth and water-soluble responses [53].

To further confirm *GH3.6* and *PP2C44* as downstream target genes of PtrABR1, the analysis of their promoter sequences revealed the presence of GCC box and DRE cis-acting elements, respectively (Appendix A). The AP2 structural domain of AP2/ERF family genes can directly bind to these elements, which are typically located in the promoter regions of stress response and tolerance genes [37,55]. Previous studies have also demonstrated that PtrABR1 in *Poncirus trifoliata* activates the transcription of PtrBAM3 by binding to the GCC box [29]. ABR1 in Arabidopsis promotes auxin production and rooting by binding to the GCC box on the *ASA1* promoter, thereby activating its expression [64]. Therefore, it is increasingly possible that *PtrABR1* enhances drought stress tolerance by repressing the expression of *GH3.6* and *PP2C44* through directly binding to their respective cis-acting elements.

## 4. Materials and Methods

### 4.1. Bioinformatics Analysis of PtrABR1

The phylogenetic tree was generated utilizing the neighbor-joining method in MEGA 7. In addition, the domain differences of the homologous genes of *P. trichocarpa* were compared using MEME analysis, and the functions of PtrABR1 were further predicted. Potential specific cis elements in the promoter region were identified through the utilization of the PlantCARE online database (http://bioinformatics.psb.ugent.be/webtools/plantcare/html/, accessed on 25 June 2021).

### 4.2. Extraction of Total RNA and RT-qPCR Evaluation

E.Z.R.A.^®^ Plant RNA Kit was used to extract the total RNA from the harvested *P. trichocarpa* plants, and TOYOBO ReverTra Ace^®^ qPCR RT Master Mix was used to reverse-transcribe the total RNA into cDNA. The manufacturer’s instructions for reactions on a 96-well fluorescent qRT-PCR instrument (Roche Light Cycler 480 II, Basel, Switzerland) were followed for conducting the qRT-PCR utilizing UltraSYBR Mixture luminous dye (CWBIO, Beijing, China). The relative abundance of transcripts was determined using the 2^−ΔΔCT^ method. Primers are shown in Appendix A.

### 4.3. Plant Materials and Drought Treatments

Plants of *P. trichocarpa* clone Donglin were cultured in vitro on a medium supplemented with 25 g L^−1^ sucrose, 5.8 g L^−1^ Gelrite, 2.41 g L^−1^ WPM, and 0.1 mg L^−1^ IBA [69]. 

Culture conditions were as follows: 25 °C, 46 μmol photons m^−2^ S^−1^, and 16 h light/8 h dark cycles. Four abiotic stresses were used on 20-day-old tissue-cultured seedlings. Selected uniformly grown histopathic seedlings were cleared of root fixation medium and placed in liquid MS medium with 200 mM NaCl and 100 mM ABA, and samples were collected at 0, 1, 3, 6, 12, and 24 h. The method refers to [70]. The histopathogenic seedlings cleared of the bottom medium were placed on clean filter paper to simulate drought, and samples were collected at 0, 0.5, 1, 3, and 6 h. The method refers to [71]. In addition to this, the seedlings were placed directly into a 4 °C refrigerator for low-temperature stress, and samples were collected at 0, 1, 3, 6, 12, and 24 h. Sample sites were obtained from leaves, snap-frozen in liquid nitrogen, and subsequently stored at −80 °C in the refrigerator for RNA extraction. 

The soil seedling drought treatment involves transplanting 4-week-old tissue culture seedlings into pots containing soil. They were grown in a greenhouse at 23–25 °C for 50 days. Cultural conditions were 16 h of light and 8 h of darkness (a long period of daylight).

### 4.4. Relative Soil Water Content of Drought Treatments

For the control of relative soil water content during drought treatment of soil-grown seedlings, we referred to [72] with modifications. Before the drought treatment, the plants in each pot were watered to saturation, at which time the soil moisture content of each pot was 100%, and the pots were weighed as a whole and recorded as m_0_. During drought stress, we weighed the pots m_1_, m_2_, etc., and calculated the relative water content using m_1_/m_0_ × 100% every day. Appropriate rehydration was performed when inconsistencies occurred to ensure consistent level of relative water content in the soil for consistent level of drought stress. The mass of the plants in the pots was negligible due to the large difference in mass between the soil and the plants in the pots.

### 4.5. Vector Construction and Plant Transformation

The CDS of *PtrABR1* and *PtrYY1* were fused with the pBI121 plant expression vector, which contained the fusion of green fluorescent protein (GFP) reporter gene fusion. *PtrABR1* promoter sequences were fused to pBI121 with a plant expression vector containing a GUS reporter gene. All the constructs were infected into *P. trichocarpa* through Agrobacterium tumefaciens-mediated transformation [69].

### 4.6. Subcellular Localization Analysis

The *PtrABR1* CDS, except for the stop codon, was amplified and subsequently inserted into the PBI121 vector containing GFP. The fusion construct (35S: PtrABR1: *GFP*) and the empty vector (35S: *GFP*) were integrated into *Agrobacterium* rhizogenes GV3101 and transferred into tobacco leaf epidermal cells using the injection method [73]. Infected tobacco was incubated in the dark for 24 h, followed by normal conditions for 48 h. Tobacco leaves were photographed with a confocal laser scanning microscope (Axio Scope A1, Carl Zeiss, Jena, Germany).

### 4.7. Nimble Cloning

Nimble Cloning (NC Cloning) enables the cloning of PCR products or DNA from introductory clones into the NC system’s cyclic expression vector via Nimble Mix. It is similar to Gateway cloning in that it does not require linearization of the expression vector. The primer design and operation are based on the Nimble Cloning kit (NC Biotech) [74].

### 4.8. Transcriptional Activation Assay

The CDS sequence of PtrABR1 and the corresponding truncated sequence were cloned and constructed into the pGBKT7 (BD) vector fused to GAL4, and the different constructs were then transformed into the Y2HGold yeast strain with the empty vector. The transformed yeast strains were cultured on SD/-Trp, SD/-Trp-His, and SD/-Trp-His + X-α-gal and incubated at 30 °C for 3–5 day. Primers are shown in Appendix A.

### 4.9. Yeast One-Hybrid (Y1H) Assay

The promoter sequence of *PtrABR1* was cloned into the pAbAi vector as bait. pBait-AbAi was then transformed into yeast strain Y1HGold, and positive colonies were obtained on SD/-Ura medium [75]. The CDS sequences of *PtrYYI*, *PtrSPL10*, and *PtrABI4* were integrated into the pGADT7 vector as prey and transformed into yeast strain Y1HGold (pBait-AbAi), respectively. Interactions were detected most in SD/-Leu medium with or without 300 ng/mL AbA. It is worth mentioning that the promoter sequence of *PtrABR1* was cloned into the pAbAi vector using the Nimble Cloning method with the primer name NC-Q-F/R, and the CDS sequences of *PtrYYI*, *PtrSPL10*, and *PtrABI4* were cloned into the pGADT7 vector using the same method with primer names NC-YY1-F/R, NC -SPL10-F/R, and NC-ABI4-F/R. All primers are shown in Appendix A.

### 4.10. Dual Luciferase Assay

The full-length CDS of *PtrYYI*, *PtrSPL10*, and *PtrABI4* and the promoter sequence of *PtrABR1* were cloned into the pGreenII 62-SK and pGreenII 0800-Luc vectors, respectively [76]. All the constructs were transformed into Agrobacterium tumefaciens GV3101-pSoup. Agrobacterium tumefaciens was suspended in an osmotic buffer solution (10 mM MES, 10 mM MgCl_2_, and 150 mM acetosyringone, at a pH of 5.6) to OD600 [74]; TFs and promoters were mixed at a *v*/*v* ratio of 10:1 and permeated into the leaves of tobacco through a needle-free syringe. The transient expression in the tobacco leaves after 3 day of penetration was analyzed using the double luciferase assay kit (Promega). The ratio of Luc to Ren activity was used to measure the regulatory effect of TFs on proPtrABR1. It is worth mentioning that the promoter sequence of *PtrABR1* was cloned into the pGreenII 0800-Luc vector using the Nimble Cloning method with the primer name NC-Q-F/R. The CDS sequences of *PtrYYI*, *PtrSPL10*, and *PtrABI4* were cloned into the pGreenII 62-SK vector by the same method with primer names NC-YY1-F/R, NC -SPL10-F/R, and NC-ABI4-F/R, respectively. All primers are shown in Appendix A.

### 4.11. GUS Staining

The full-length CDS of *PtrYYI*, *PtrSPL10*, and *PtrABI4* and the promoter sequence of *PtrABR1* were constructed into pGreenII 62-SK and pBI121-GUS vectors, respectively. The CDS sequences of *PtrYYI*, *PtrSPL10*, and *PtrABI4* were constructed into pGreenII 62-SK vectors using the Nimble Cloning method with primers named NC-YY1-F/R, NC -SPL10-F/R, and NC-ABI4-F/R, respectively. In addition, the promoter sequence of *PtrABR1* was constructed into pBI121-*GUS* vector using Vazyme ClonExpress II One Step Cloning Kit with primer pBI121-ProABR1-F/R. All constructs were transformed into *Agrobacterium* GV3101-pSoup receptor cells. The promoter and transcription factor Agrobacterium suspension culture was mixed and infiltrated into tobacco leaves. After 48 h of incubation in an incubator, tobacco leaves were collected for GUS staining and quantitative GUS gene analysis [77]. All primers are shown in Appendix A.

Histochemical staining was conducted using 5-bromo-4-chloro-3-indolyl-β-D-glucuronide cyclohexyl ammonium salt (X-gluc), following the previously described protocols [78,79]. Briefly, ProPtrABR1::*GUS* transgenic poplar was immersed in GUS assay solution for the corresponding time, vacuum permeabilized for 2 h, and left overnight at 37 °C in the dark. The stained poplar material was decolorized in a decolorization solution for 6–8 h. After decolorization, it was observed and photographed under an Olympus ZX7 microscope (Olympus Corporation, Tokyo, Japan).

### 4.12. RNA Sequencing and Analysis

To perform RNA sequencing (RNA-Seq) analysis, three WT and three transgenic plants were selected at the age of 20 day, and their roots were collected to generate six samples; each sample was taken from six plants. RNA-Seq experiments and bioinformatics analyses were carried out by ANOROAD (Beijing, China).

### 4.13. Determination of Relevant Physiological Indicators under Drought Stress

The dry weight, plant height, electrolyte leakage (EL), and malondialdehyde (MDA) of roots were determined after 7 day of growth in the rooting medium and 14 day of growth in medium with or without 6% PEG6000.

Soil-cultured seedlings were selected for consistent growth over 20 day and moved to the soil for planting. Drought treatment was applied at 50 day. The measurement methods of root fresh weight, dry weight, EL, MDA [38,80], and Pro content [81] were all derived from previous experimental studies.

### 4.14. Statistical Analysis

We used GraphPad Prism 7 software for statistical analysis and visualization of the sample data. All the experiments were repeated independently at least three times, and the differences of measured parameters were statistically significant tested by Student’s *t*-test; there was a significant difference at * *p* < 0.05, ** *p* < 0.01, and *** *p* < 0.001.

## 5. Conclusions

To sum up, *PtrABR1* improved the drought tolerance of *P. trichocarpa* by promoting the growth and development of lateral roots. PtrYY1 and PtrSPL10 could directly activate and repress the promoter activity of *PtrABR1*, respectively. The further exploration of the function of PtrYY1 revealed that *PtrYY1* also improved the drought tolerance of *P. trichocarpa* by promoting the growth and development of lateral roots. In addition, by analyzing the RNA-seq and RT-qPCR results, the potential downstream target genes of PtrABR1, *PP2C44*, and *GH3.6* were predicted, and it was hypothesized that *PtrABR1* might regulate root growth and anti-drought by suppressing the expression of *PP2C44* and *GH3.6*. PtrABR1-mediated drought tolerance would contribute to breeding anti-drought poplar tree species which have higher economic and ecological values. It also provides new insights into the drought response mechanism of plant roots.

## Figures and Tables

**Figure 1 ijms-24-13748-f001:**
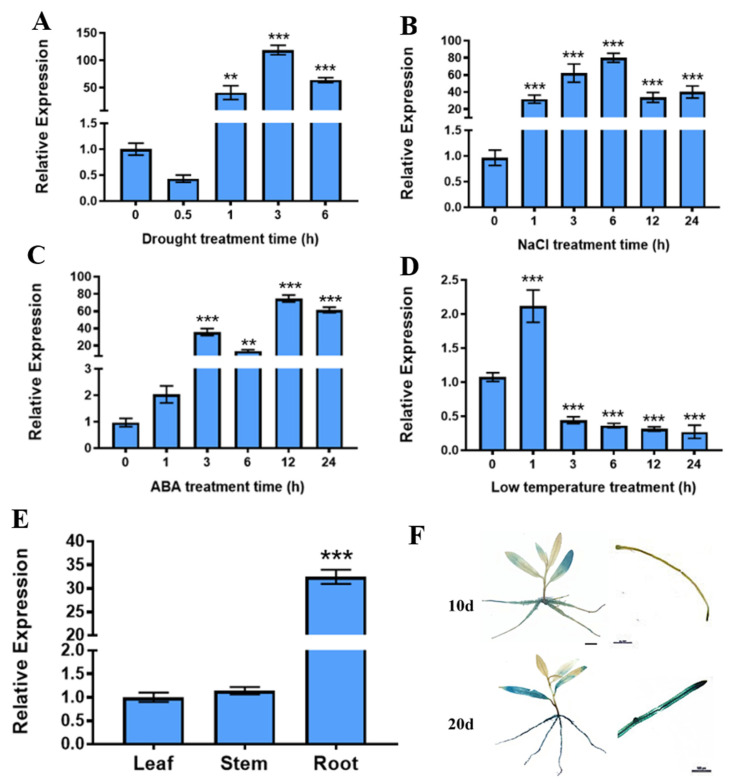
Expression analysis of *PtrABR1* under abiotic stress and tissue specificity. RT-qPCR analysis of PtrABR1 under drought (**A**), NaCl (**B**), ABA (**C**), and low temperature (**D**). (**E**) Tissue-specific expression of *PtrABR1*. (**F**) GUS staining of PtrABR1pro::*GUS* transgenic plants at day 10 and day 20, Bar = 500 μm. Student’s *t*-tests, ** (*p* < 0.01), and *** (*p* < 0.001).

**Figure 2 ijms-24-13748-f002:**
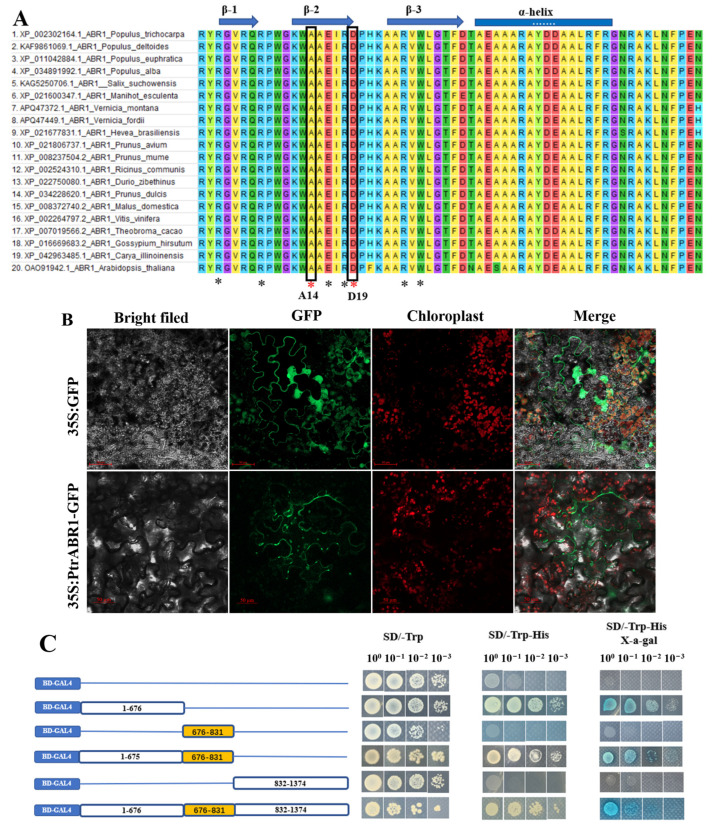
Characterization of the PtrABR1 gene in *P. trichocarpa*. (**A**) Multiple comparisons of ABR1 from other plants with the AP2 structural domain of PtrABR1. Identical colors indicate identical amino acid residues. Black asterisks indicate DNA-binding residues, while red asterisks indicate two characteristic residues of the ERF subfamily. Arrows and bars indicate the β-helix and α-helix regions, respectively. (**B**) Localization of PtrABR1 protein in tobacco cells. Based on visualization of green fluorescent protein (GFP) in tobacco leaves transformed with a fusion construct (35S: PtrABR1-GFP) or empty vector (35S: GFP). gfp, green fluorescence; red, chloroplast autofluorescence; scale bar = 50 μm. (**C**) Validation of PtrABR1 transcriptional activation activity. Schematic diagrams are for the null, N-terminal, AP2 structural domain, N-terminal and AP2 structural domains, C-terminal, and full length. The numbers above the bars indicate the position of amino acid residues.

**Figure 3 ijms-24-13748-f003:**
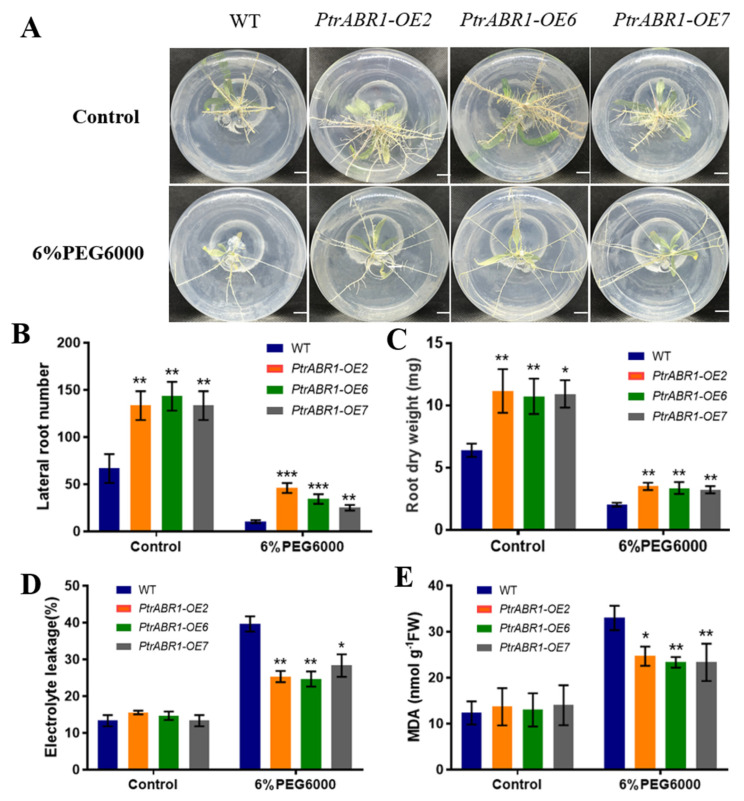
*PtrABR1* positively regulates the drought tolerance of poplar roots. (**A**) Root morphology of *PtrABR1-OE* and WT plants treated with 6% PEG6000 and left untreated. Bar, 1 cm. (**B**) The number of lateral roots, (**C**) root dry weight, (**D**) EL, and (**E**) MDA content. Student’s *t*-tests, * (*p* < 0.05), ** (*p* < 0.01), or *** (*p* < 0.001).

**Figure 4 ijms-24-13748-f004:**
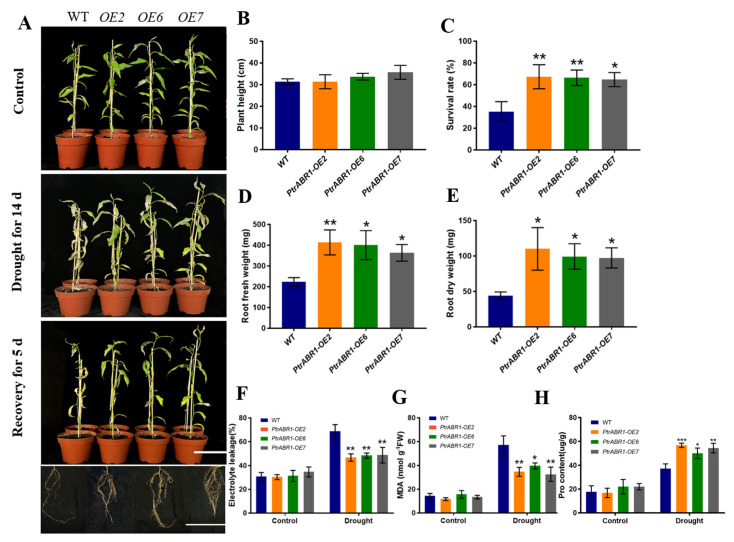
*PtrABR1* improves drought tolerance by promoting lateral root growth. (**A**) Morphological characteristics of in vitro cultured 4-week-old poplar plants grown in pots for 50 day, subjected to drought for 14 day, and then watered for 5 day. Bar, 10 cm. Plant height (**B**), survival rate (**C**), root fresh weight (**D**), root dry weight (**E**), electrolyte leakage (EL) (**F**), MDA content (**G**), and Pro content (**H**). Student’s *t*-tests, * (*p* < 0.05), ** (*p* < 0.01), or *** (*p* < 0.001).

**Figure 5 ijms-24-13748-f005:**
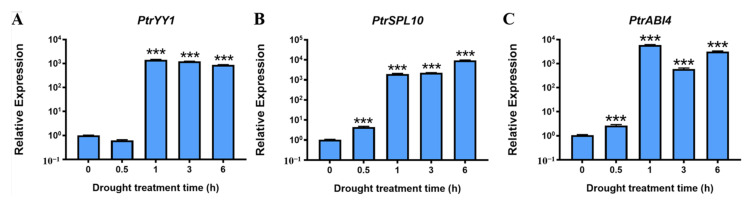
*PtrYY1*, *PtrSPL10*, and *PtrABI4* respond to drought stress and are mainly expressed in roots. (**A**–**C**) Expression of *PtrYY1*, *PtrSPL10*, and *PtrABI4* in roots after 0, 0.5, 1, 3, and 6 h of drought treatment. Plotting vertical coordinates on a log10 interval scale. Student’s *t*-tests, *** (*p* < 0.001).

**Figure 6 ijms-24-13748-f006:**
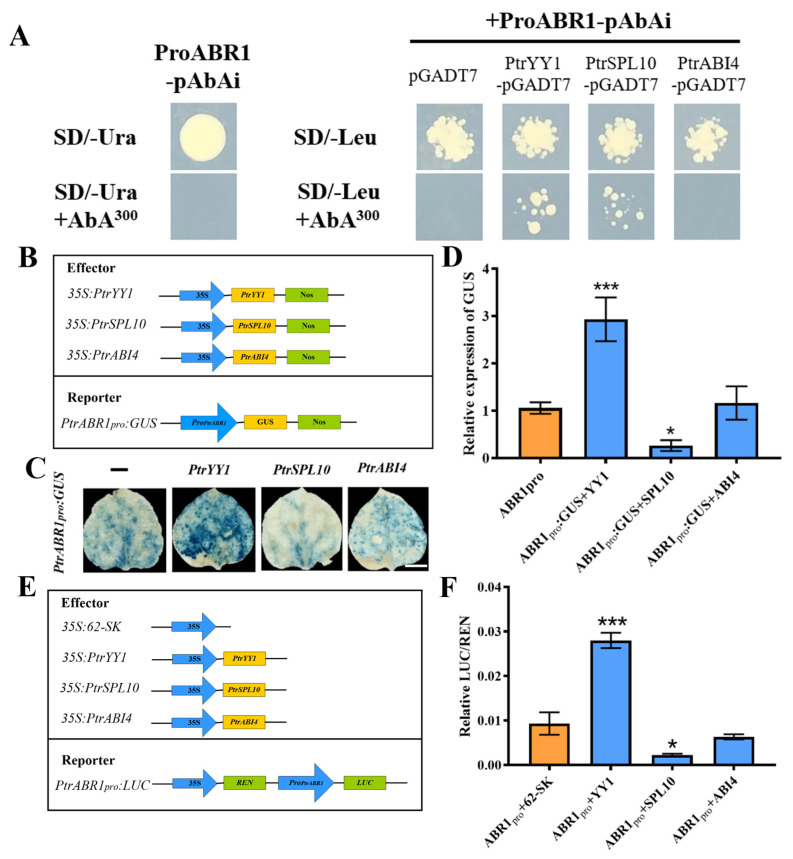
Regulation of PtrYY1, PtrSPL10, and PtrABI4 on the *PtrABR1* promoter. (**A**) Y1H analysis. In the presence of AbA (SD/-Leu+AbA), the binding activity was determined on the SD medium lacking Leu. (**B**) Schematic diagram of GUS reporter gene, reporter, and effector detected by textile chemical staining. (**C**) A schematic diagram of GUS reports of gene activity detected by histochemical staining; blue indicates GUS activity. Bar = 1 cm. (**D**) GUS reports of the relative expression of genes. (**E**) dual luciferase assays, reporter, and effector. (**F**) Relative luciferase activity in tobacco leaves determined from dual luciferase reporter. Student’s *t*-tests, * (*p* < 0.05) and *** (*p* < 0.001).

**Figure 7 ijms-24-13748-f007:**
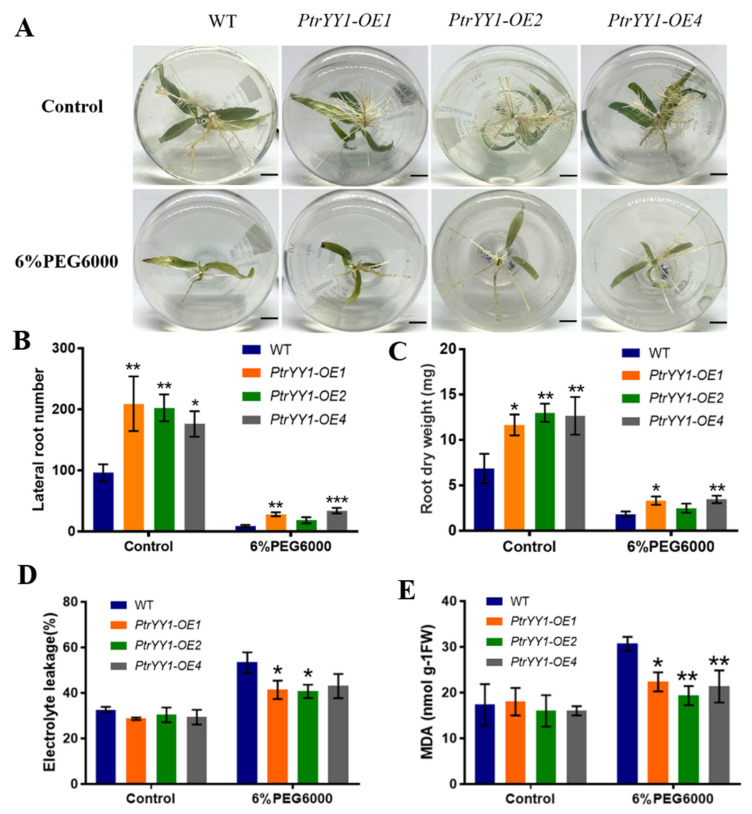
*PtrYY1* positively regulates the drought tolerance of poplar roots. (**A**) Root morphology of *PtrYY1-OE* and WT plants treated with 6% PEG6000 and left untreated. Bar, 1 cm. (**B**) The number of lateral roots, (**C**) root dry weight, (**D**) EL, and (**E**) MDA content. Student’s *t*-tests, * (*p* < 0.05), ** (*p* < 0.01), or *** (*p* < 0.001).

**Figure 8 ijms-24-13748-f008:**
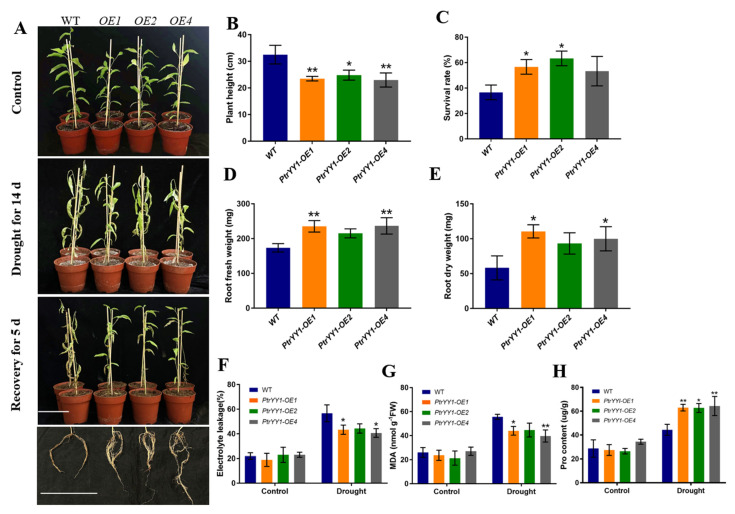
*PtrYY1* improves drought tolerance by promoting lateral root growth. (**A**) Morphological characteristics of in vitro cultured 4-week-old poplar plants grown in pots for 50 day, subjected to 14 day of drought and then 5 day of watering. Bar, 10 cm. Plant height (**B**), survival rate (**C**), root fresh weight (**D**), root dry weight (**E**), electrolyte leakage (EL) (**F**), MDA content (**G**), and Pro content (**H**). Student’s *t*-tests, * (*p* < 0.05) and ** (*p* < 0.01).

**Figure 9 ijms-24-13748-f009:**
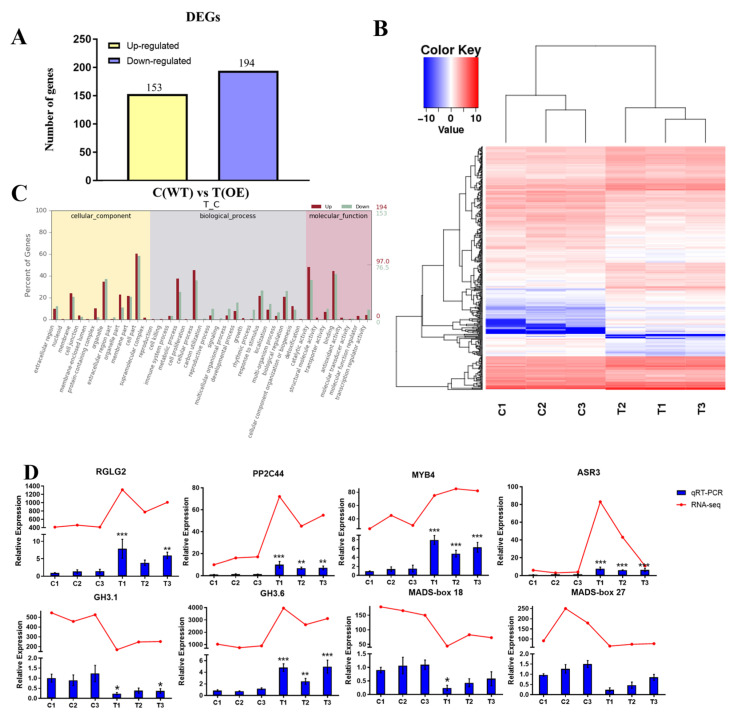
Transcriptome analysis of *PtrABR1* transgenic plants and WT plants. (**A**) Histogram analysis of the number of DEGs in the roots of WT and *PtrABR1-OE* plants. (**B**) Heat map of differentially expressed genes. (**C**) GO analysis of differentially expressed genes (DEGs). DEGs are annotated based on biological processes, molecular function, and cellular composition. (**D**) Validation of 8 DEGs expression levels via qRT-PCR. Student’s *t*-tests, * (*p* < 0.05), ** (*p* < 0.01), or *** (*p* < 0.001).

**Figure 10 ijms-24-13748-f010:**
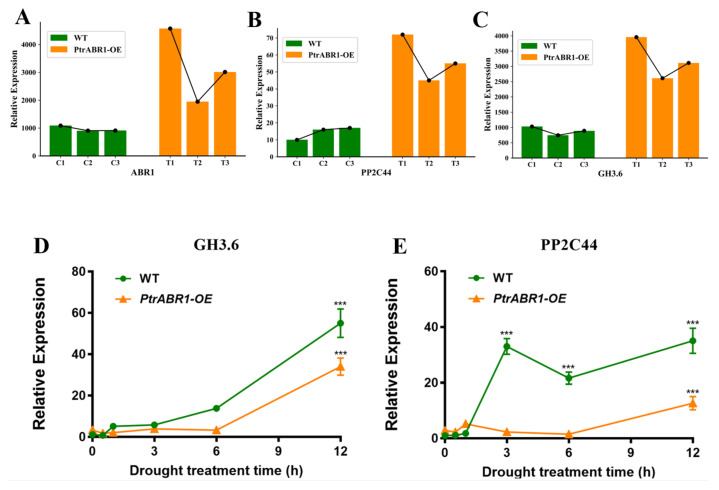
Prediction of downstream target genes of PtrABR1 under drought stress. (**A**–**C**) The RNA-seq results of *GH3.6* and *PP2C44* expression in different WT and *PtrABR1-OE* lines. (**D**,**E**) WT and *PtrABR1-OE* plants that were 20 day old were placed on filter paper and exposed to drought for 0, 0.5, 1, 3, 6, and 12 h. Roots were collected at different times for qRT-PCR analysis of *GH3.6* and *PP2C44*, respectively. Student’s *t*-tests, *** (*p* < 0.001).

**Figure 11 ijms-24-13748-f011:**
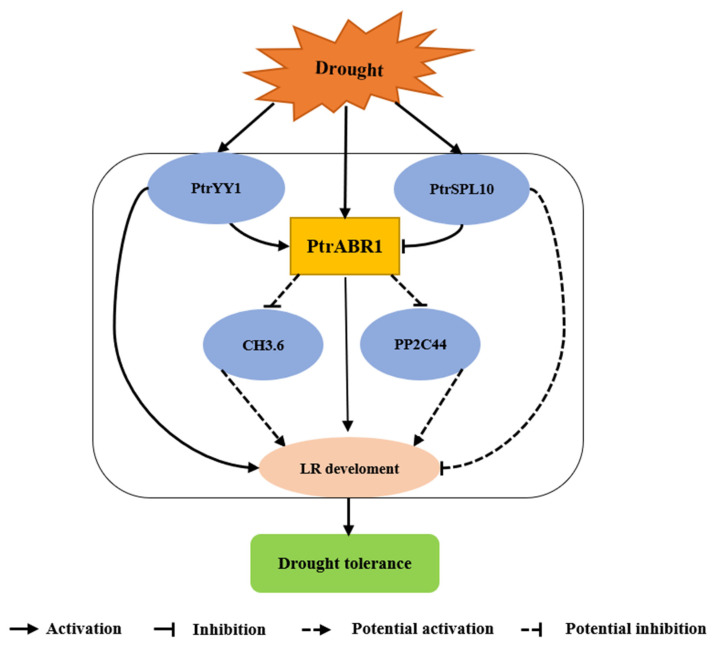
*PtrABR1* is a model aimed at enhancing drought resistance by promoting the growth of lateral roots. This is achieved through its direct induction by *PtrYY1*, which enhances tolerance to drought stress by fostering the growth and development of lateral roots. Furthermore, it is speculated that PtrABR1 also directly inhibits the expression of *GH3.6* and *PP2C*, thereby further improving poplar drought resistance.

## Data Availability

The data supporting the conclusions of this manuscript have been displayed as figures and Appendix A and will be made available by the authors, without undue reservation, to any qualified researcher.

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
