# Peer review of "PtrABR1 Increases Tolerance to Drought Stress by Enhancing Lateral Root Formation in Populus trichocarpa"

_ijms, 2023, doi:10.3390/ijms241813748_

Round 1

Reviewer 1 Report

The manuscript 'PtrABR1 increases tolerance to drought stress by enhancing lateral root formation in Populus trichocarpa is looks interesting. I have some suggestions regarding this :

1) In Introduction part, It will be good if you can add more reports on ABR1, YY1 and ABA responses in relation to drought stress.

2) In Figure.2 (B) confocal image of 35S:PtrABR1-GFP showing very less expression? It will be better if you can explain this or provide a different image.

3) In Figure.6 (B,C,D,E,F), Gus and luciferase legends and the figures are not correlating. Please recheck.

4) In Promoter Gus assay, how you transformed constructs PtrYY1/SPL10/ABI4. please mention in materials and methods section.

5) Have you checked gene copy number of your transgenics?

6) In Discussion, it will be nice if you can elaborate PtrYY1-PtrABR1-ABA responses in drought stress. Because there are reports which were mentioned ABR1 and YY1 act negatively on ABA and we know ABA has a major role in drought. So, how it can be fit with your findings?

7) In supplementary, please mention the size of your marker and gene in gel pioctures.

Author Response

Point 1: In Introduction part, It will be good if you can add more reports on ABR1, YY1 and ABA responses in relation to drought stress.

Response 1: Thank you very much for your valuable comments, we have supplemented more ABR1, YY1 and ABA responses in relation to drought stress in the Introduction section. The additions you will find in the new manuscript.

Point 2: In Figure.2 (B) confocal image of 35S:PtrABR1-GFP showing very less expression? It will be better if you can explain this or provide a different image.

Response 2: We appreciate it very much for this good suggestion. We have thought carefully about this question and there are the following possible reasons why the images show low expression:

Cell specificity: The gene may be expressed only in a specific cell type or a specific developmental stage and less in other cell types or developmental stages. This results in the gene being detected with low expression when the gene is localized at the subcellular level.

In addition, considering the problematic image clarity in the previous version of the manuscript, we have replaced the clear figure in the new manuscript

Point 3: In Figure.6 (B,C,D,E,F), Gus and luciferase legends and the figures are not correlating. Please recheck.

Response 3: We are very sorry for our errors and corrected it. Thanks so much for your useful comments.

Point 4: In Promoter Gus assay, how you transformed constructs PtrYY1/SPL10/ABI4. please mention in materials and methods section.

Response 4: We appreciate it very much for this good suggestion, and we have supplemented the experimental method accordingly.

Point 5: Have you checked gene copy number of your transgenics?

Response 5: Thank you very much for your valuable comments. After obtaining the transgenic line, I checked the gene copy number. Among them, the promoter transgenic line is in Figure S2B, the PtrABR1 transgenic line is in Figure S3B, and the PtrYY1 transgenic line is in Figure S6C.

Point 6: In Discussion, it will be nice if you can elaborate PtrYY1-PtrABR1-ABA responses in drought stress. Because there are reports which were mentioned ABR1 and YY1 act negatively on ABA and we know ABA has a major role in drought. So, how it can be fit with your findings?

Response 6: Thank you very much for your valuable comments, we have supplemented them in the Discussion section. The additions will be found in the new manuscript.

Point 7: In supplementary, please mention the size of your marker and gene in gel pioctures.

Response 7: We are very grateful for this good suggestion, and we have supplemented the length of the gene fragments in the gel images and made a note of the DNA marker in the legend.

Reviewer 2 Report

The manuscript entitled “PtrABR1 increases tolerance to drought stress by enhancing lateral root formation in Populus trichocarpa” is well-written and organised properly. I enjoyed reading the whole of the manuscript. The study is Informative and attempts to discover new information on a certain topic. However, a few major questions need to be addressed before publishing in reputed journal.

 Why, the authors have not used knockout with overexpressor to investigate the exact impact of PtrABR1?

The authors have not included parameters such as proline, which is an important biomarker for analyzing drought; why?

 The conclusion needs to be elaborate. Authors are suggested to consider Figure 11 signal cascade to conclude the hypothesis.

 References should be in a single format; for example, 1-34 references are in a different format, and 35-38 are in a different format.  

Author Response

Point 1:  Why, the authors have not used knockout with overexpressor to investigate the exact impact of PtrABR1?

Response 1: Thank you very much for your valuable comments. We have also constructed expression vectors using CRISPR gene editing technology and performed genetic transformation several times, but unfortunately each time the positive plants were sequenced and identified, we found that they were not edited. But fortunately we obtained multiple overexpression lines and found significant phenotypic and drought resistance compared to WT. Regarding the non-obtaining of knockout transgenic plants, our laboratory will continue with the aim of further validating the results of this experiment. Thank you again for your valuable suggestions.

Point 2: The authors have not included parameters such as proline, which is an important biomarker for analyzing drought; why?

Response 2: Thank you very much for your valuable comments. We measured the proline content before and after drought when we drought-treated the soil-grown seedlings, but when selecting the data, I did not choose to attach the figure of proline in order to have a slightly more aesthetically pleasing symmetry in the layout, and I apologize for this wrong choice. The proline figure has been supplemented to the new manuscript, and we thank you again for your valuable comments.

Point 3: The conclusion needs to be elaborate. Authors are suggested to consider Figure 11 signal cascade to conclude the hypothesis.

Response 3: We are very grateful for this good suggestion, and we have redescribed it in the conclusion section concerning Figure 11.

Point 4:  References should be in a single format; for example, 1-34 references are in a different format, and 35-38 are in a different format.  

Response 4: Thank you very much for your valuable comments, we carefully checked the manuscript and made corrections.

Reviewer 3 Report

The article «PtrABR1 increases tolerance to drought stress by enhancing lateral root formation in Populus trichocarpa» written by Lijiao Sun, Xinxin Dong and Xingshun Song is devoted to the study of the molecular mechanisms of black cottonwood adaptation to drought. This occurs due to the regulation of the upstream (PtrYY1) and downstream (PtrGH3.6 and PtrPP2C44) target genes by the PtrABR1 gene. At the level of the whole organism, this leads to the growth and development of roots, improving the resistance of plants to stress. The article is of undoubted scientific interest. It contains its own original data.

The article has a few comments:

1) Remove the dot after the title of the article.

2) Carefully check throughout the article text: the Latin names of species and the names of genes should be italicized.

3) Carefully check throughout the text of the article: in some places, there are no gaps in the sentences.

4) In the introduction, the authors refer to an article on Poncirus trifoliate. Due to the identical first letters of the species and generic names of black cottonwood and trifoliate orange, the Latin species abbreviated names (Ptr) located before the genes are identical for these species. This may cause confusion in future publications and databases.

5) Figure 1, line 113: replace "drought, ABA, NaCl, and low temperature" with "(A) drought, (B) NaCl, (C) ABA, and (D) low temperature" as the figures situated in such a sequence.

6) Line 113: is it double colon right?

7) What is the reason for using the Student's test when testing statistical hypotheses? Specify the statistical programs in which data analysis was carried out. Indicate in which program the data visualization was performed (biochemistry, physiology, gene expression etc.).

8) The data in Figure 5 should be displayed on a logarithmic scale.

9) There is no characteristic of the object of study.

10) How many plants took part in physiological and biochemical experiments?

11) Lines 551-552: The sentence has no ending.

12) Line 760: replace 7880 with 78.

Author Response

Point 1:  Remove the dot after the title of the article.

Response 1: Thank you very much for your valuable comments, and we have removed the dot.

Point 2: Carefully check throughout the article text: the Latin names of species and the names of genes should be italicized.

Response 2: We are very grateful for this good suggestion, and we carefully checked the manuscript and made corrections.

Point 3: Carefully check throughout the text of the article: in some places, there are no gaps in the sentences.

Response 3: Thank you very much for your valuable comments, we carefully checked the manuscript and made corrections.

Point 4:  In the introduction, the authors refer to an article on Poncirus trifoliate. Due to the identical first letters of the species and generic names of black cottonwood and trifoliate orange, the Latin species abbreviated names (Ptr) located before the genes are identical for these species. This may cause confusion in future publications and databases.

Response 4: Thank you very much for your valuable comments as well, and I apologize for causing such confusion.

First of all, there are two other studies involving PtrABR1 in our group that are under external review, one of which is about to be published, so if we make the correction now, it may result in an inconsistency in the name of the PtrABR1 gene in that species.

In addition, PtrABR1 in this study and PtrABR1 in Poncirus trifoliate are described in detail concerning the species to which they belong, and the species is indicated in the title of this study, which is believed to make it easier to distinguish them.

Finally, the use of (Ptr) is because of the reference to the abbreviation of other genes in Populus trichocarpa (Chen et al., 2013; Helariutta & Kucukoglu Topcu, 2023; Karim et al., 2015; Peng et al., 2019). Our pre-consideration is that we hope it can be consistent with the abbreviations of other genes of Populus trichocarpa.

Thank you again for your valuable suggestion, and I will avoid such a problem in the future.

Point 5: Figure 1, line 113: replace "drought, ABA, NaCl, and low temperature" with "(A) drought, (B) NaCl, (C) ABA, and (D) low temperature" as the figures situated in such a sequence.

Response 5: We are very sorry for our errors and corrected them. Thanks so much for your useful comments.

Point 6: Line 113: is it double colon right?

Response 6: We are very sorry for our errors and corrected it in the new manuscript.

Point 7: What is the reason for using the Student's test when testing statistical hypotheses? Specify the statistical programs in which data analysis was carried out. Indicate in which program the data visualization was performed (biochemistry, physiology, gene expression etc.).

Response 7: Thank you very much for your question. t-test is a method used to compare the difference between the means of two groups for small sample situations. It determines whether there is a significant difference between the means of two groups of samples based on the relationship between the sample mean, the sample variance, and the sample size, hence the use of this test. The program used for statistical analysis and visualization of the sample data is supplemented in the methods section. Thanks again for your valuable suggestions.

Point 8: The data in Figure 5 should be displayed on a logarithmic scale.

Response 8: Thank you for your valuable suggestion, we have made changes to Figure 5.

Point 9: There is no characteristic of the object of study.

Response 9: I'm very sorry that I didn't quite understand what you meant, but I'm guessing that by the lack of characterization, you are referring to the above-ground phenotype of the transgenic plants.

First of all, thank you very much for your valuable input, although the above-ground part of the transgenic plant is not characterized, it is fortunate to find that it affects the formation of lateral roots and plays an important role in drought stress. Thank you again for your valuable comments.

Point 10: How many plants took part in physiological and biochemical experiments?

Response 10: Thank you very much for your question. Before performing the treatments, we selected 10 plants from each line took part in participate in the physiological and biochemical experiments.

Point 11: Lines 551-552: The sentence has no ending.

Response 11: We are very sorry for our errors and have supplemented it in the new manuscript.

Point 12: Line 760: replace 7880 with 78.

Response 12: We are very sorry for our errors and corrected it in the new manuscript.

References

Chen, Y., Yang, J., Wang, Z., Zhang, H., Mao, X., & Li, C. (2013). Gene structures, classification, and expression models of the DREB transcription factor subfamily in Populus trichocarpa. The Scientific World Journal, 954640. https://doi.org/10.1155/2013/954640

Helariutta, Y., & Kucukoglu Topcu, M. (2023). Epigenetics rules cambial growth. Nature plants, 9(1), 7–8. https://doi.org/10.1038/s41477-022-01316-6

Karim, A., Jiang, Y., Guo, L., Ling, Z., Ye, S., Duan, Y., Li, C., & Luo, K. (2015). Isolation and characterization of a subgroup IIa WRKY transcription factor PtrWRKY40 from Populus trichocarpa. Tree physiology, 35(10), 1129–1139. https://doi.org/10.1093/treephys/tpv084

Peng, X., Pang, H., Abbas, M., Yan, X., Dai, X., Li, Y., & Li, Q. (2019). Characterization of Cellulose synthase-like D (CSLD) family revealed the involvement of PtrCslD5 in root hair formation in Populus trichocarpa. Scientific Reports, 9(1), 1452. https://doi.org/10.1038/s41598-018-36529-3

Reviewer 4 Report

Dear authors,

Conclusion is too short and does not describe the overall story of the manuscript. I think you should write with scientific findings and future message also for upcoming research in the field. Rest of the paper is written very well.

Author Response

Point 1:  Conclusion is too short and does not describe the overall story of the manuscript. I think you should write with scientific findings and future message also for upcoming research in the field. Rest of the paper is written very well.

Response 1: Thank you very much for your valuable comments. We have redescribed the conclusion section in the new manuscript. Once again, thank you for your suggestion, which is significant for the enhancement of the manuscript's content.
